# An Efficient Resource Allocation Strategy for Edge-Computing Based Environmental Monitoring System

**DOI:** 10.3390/s20216125

**Published:** 2020-10-28

**Authors:** Juan Fang, Juntao Hu, Jianhua Wei, Tong Liu, Bo Wang

**Affiliations:** 1Faculty of Information Technology, Beijing University of Technology, Beijing 100124, China; hujuntaochn@emails.bjut.edu.cn (J.H.); weijianhua@bjut.edu.cn (J.W.); 2Beijing Computing Center, Beijing 100094, China; liutong@bcc.ac.cn; 3National Computer Network Emergency Response Technical Team, Coordination Center of China, Beijing 100096, China; wb@cert.org.cn

**Keywords:** environmental monitoring, edge computing, resource allocation, task scheduling

## Abstract

The cloud computing and microsensor technology has greatly changed environmental monitoring, but it is difficult for cloud-computing based monitoring system to meet the computation demand of smaller monitoring granularity and increasing monitoring applications. As a novel computing paradigm, edge computing deals with this problem by deploying resource on edge network. However, the particularity of environmental monitoring applications is ignored by most previous studies. In this paper, we proposed a resource allocation algorithm and a task scheduling strategy to reduce the average completion latency of environmental monitoring application, when considering the characteristic of environmental monitoring system and dependency among task. Simulations are conducted, and the results show that compared with the traditional algorithms. With considering the emergency task, the proposed methods decrease the average completion latency by 21.6% in the best scenario.

## 1. Introduction

With the development of industry and the acceleration of urbanization, we must pay attention to environmental monitoring, because air quality has a paramount impact on human health. The collection and processing of environmental data are prerequisites for environmental monitoring and pollution warning. As a novel technology, the Internet of Things (IoT) is considered to be a pollution monitoring solution, which attracts attention from both academia and industry. In addition, cloud computing makes up the lack of computing resources and energy capacity in environmental monitoring terminals devices. In an environmental monitoring system based on IoT and cloud computing technology, the environmental data collected by the sensors are simply processed by the IoT devices and uploaded to the cloud server. Subsequently, the cloud server processes and analyzes the environmental data uploaded from all sensors. However, with the decrease of granularity of environmental monitoring and the increase of data analysis application amount, the centralized cloud computing architecture is facing challenges, such as high latency, low coverage, and lagged data transmission [1].

Edge computing, as a new computational paradigms, reduces the application completion latency and the energy consumption of data transmission by distributing cloud resources closer to where data generation [2]. Fang et al. [3] has proven that edge computing could enhance the real-time performance of service completion, with reducing computing load and power consumption in the cloud, by offloading computing request to edge servers closer to the IoT device for execution. The authors in [4] proposed a generalized logical sphere(GLS) modeling scheme in order to avoid servers overload.

In edge-computing based environmental monitoring system, sensors collect the environmental data in real time and transmit them to the edge computing server in order to execute necessary processing and analysis, which reduces system energy consumption and network traffic [5]. The accuracy of environmental monitoring is affected by positioning problems of monitoring sensors. To solve this problem, Fei et al. [6] investigated advanced parameter prediction skills and proposed a a smart collaborative tracking scheme to improve particle filter approaches. A number of previous works have proven that resource allocation strategies could reduce task completion latency and energy consumption [7,8,9]. Fengxian et al. [10] design a genetic algorithm and particle swarm optimization-based algorithm to solve resource allocation problem. However, the dependency between subtasks was not considered in this study. Non-dominated sorting genetic algorithm II was adopted in order to realize multi-objective optimization to reduce the execution time and energy consumption of edge computing devices in paper [11]. Songtao et al. [12] has proven that resource allocation policy is determined by the computing workload of a task and the maximum completion time of its immediate predecessors. In [13], a smart collaborative automation (SCA) scheme was proposed in order to improve resource usage. Du et al. [14] proposed an algorithm, which obtained allocation decision via semidefinite relaxation and randomization, but the communication among subtasks are ignored by this work. The authors in [15] took dependencies among subtasks into account, and proposed an multistage greedy adjustment (MSGA) algorithm to solve the task allocation problem. They minimized the completion time of application by jointly considering the network flows and tasks. Laizhong et al. [16] focus on the task offloading problem in order to find out the optimal tradeoff between task completion latency and energy consumption. They proposed a modified fast and elitist nondominated sorting genetic algorithm to solve the offloading problem. However, in this work, the tasks are considered to be undivided, which wastes the parallel computing capability of edge computing servers. Authors in [17] created un-excuted task queue at the MEC server and proposed an online algorithm to allocate resources. Pereira et al. [18] propose an allocation and management resources mechanism to reduce the model complexity of resource allocation algorithm. The authors in [19] take the dependency between task into account, and proposed a deep reinforcement learning approach to make offloading decision, the difference among tasks is ignored by this work. Xu et al. [20] adopt Non-dominated Sorting Genetic Algorithm II to shorten the resource allocating time of the computing tasks and reduce the energy consumption of the edge computing servers, but this work did not consider the dependencies among tasks.

In most previous studies, the particularity of environmental monitoring applications has not been considered. For example, air pollution tracing analysis can only be performed on certain servers with wind direction information of the region. In addition, as the chromosome size increases, the time complexity of the resource allocation algorithm increases exponentially [10]. In order to reduce the solution space of genetic algorithm, we cluster subtasks in genetic algorithm by k-means algorithm, which makes the resource allocation algorithm converge faster.

This study targets the minimum time cost of task completion in environmental monitoring system by fully utilizing the parallel computing capacity of edge computing. The main contributions of this work are outlined, as follows:We first introduce the computing model of task with dependency and formulate resource allocation problem. Taking the communication among subtasks into account, we propose a resource allocation algorithm based on GA. The proposed algorithm can reach a better sollution than GA under a same generation, by clustering the subtasks with heavy dependency and decreasing the size of sollution space.The impact of dependency among subtasks on completion time has been discussed in the contribution above. To solve this problem, a task scheduling strategy proposed that reordering the task queue on edge computing server according to the priority of subtasks can reduce the average task completion latency when considering emergency task.

The remainder of this paper is organized, as follows. Section 2 introduces the system model and formulates the resource allocation problem. Section 3 introduces the proposed resource allocation algorithm and task scheduling strategy. Section 4 describes a simulation analysis of the proposed algorithm and compares it with the traditional algorithm. Section 5 concludes the study.

## 2. System and Computation Model

This section describes the system model of edge-computing based environmental monitoring system and formulates the resources allocation problem.

### 2.1. System Model

In the traditional centralized cloud computing system, the data procession is mainly on cloud computing server. IoT devices collect environmental data periodically and transmit it to the edge gateways and cloud server. After processing on cloud, the results are returned to end devices. By deploying resources (i.e., computing resource and storage resource) on edge gateways, they can process some data locally, then return the processing results to IoT devices directly or transmit data to cloud for further processing.

In this work, we extend the edge-cloud heterogeneous collaborative network architecture in [3], which reduces the network bandwidth occupation, as shown in Figure 1. This network architecture consists of four parts: cloud layer, region layer, roadside layer, and IoT device layer. The devices in IoT layer collect environmental data and upload task to the directly connected roadside server in roadside layer. The roadside servers process data collaboratively or transmit them to a regional server according to resource allocation policy. Regional servers has more computing resources than roadside servers and it is responsible for executing regional tasks. When the devices in the edge network are overloaded, the task will be uploaded to cloud server. After completing all of the tasks uploaded from IoT devices, partial data will recored into databases on cloud server. Fei et al. [21] proposed a smart collaborative distribution scheme to solve the data privacy-preserving problem in such a distribution system.

We assume that there are *E* environmental monitoring sensors that are located in a region, denoted by a set of E={1,2,3,…,E}, which are installed on roadside or indoors to collect environmental data (i.e., PM2.5, PM10, SO2, NO, CO, and temperature, etc.) and offload environmental monitoring task into the edge computing system.

For each monitoring region, S represents a set of computing resources, including a set of roadside edge computing servers $S_{road}$, a set of regional servers $S_{region}$, and a cloud computing server $S_{cloud}$ as shown in Figure 1. Each server $S_{i}$ has its own resource (computing resource, bandwidth, and device layer) can be represented, as follow:(1)$$Res_{i}=\left\{f_{i},Bw_{i},flag_{i}\right\},$$
where $flag_{i}$ indicates which layer the server $S_{i}$ located on. Which can be denoted as follow:(2)$$flag_{i}=\left\{\begin{array}{cc} 0 & \mathrm{if}\;S_{i}\in S_{road},\\ 1 & \mathrm{if}\;S_{i}\in S_{region},\\ 2 & \mathrm{if}\;S_{i}\in S_{cloud}. \end{array}\right.$$

The virtual machines adopts time sharing policy, and execute various task circularly. A space sharing strategy was adopted in order to task processing mechanism for each virtual machine. Before being executed, the arriving task waits in the waiting queue as shown in Figure 2.

### 2.2. Application Model

We utilize a directed acyclic graph A=(M,D) to describe the dependency relationship among these subtasks. *M* represents the modules of an environmental monitoring application, and *D* indicates the data dependency between two modules. Each module $m_{i}\in M$ has its own resource requirement (computing resource requirements, bandwidth requirement, and minimum processing layer), which can be denoted as follows:(3)$$m_{i}=\left\{l_{i},Bw_{i},layer_{i}\right\},$$
where $layer_{i}\in \left\{0,1,2\right\}$ indicates module $m_{i}$ can only be processed on which layer or above.

Each dependency $d_{i}\in D$ can be denoted as $d_{i}=\left\{m_{src},m_{dest},data_{i}\right\}$, which describe that the processing result with $data_{i}$ bytes of $m_{src}$ is the input data of $m_{dest}$. Each $m_{i}\in M$ represents a subtask and a dependency $d_{j}=\left\{m_{i},m_{k},data_{j}\right\}$ indicates the precedence constraint between subtask $m_{i}$ and $m_{k}$, such that subtask $m_{k}$ cannot start execution until $m_{i}$ executed.

### 2.3. Latency Model

We let $f_{n,m}$ denote the computation resource of server $S_{n}$ which is allocated to subtask *m*. The execution latency of subtask *m* is given by:(4)$$T_{n,m}^{exc}=l_{m}f_{n,m}^{-1}$$

Let $T_{m,n,w}^{trans}$ denote the subtask *m* transmission latency from server *n* to *w*. Subsequently, we discuss the impact of dependency among subtasks on resource allocation algorithm. The dependencies among subtasks constraining a subtask cannot be processed until all of its predecessors have already been executed. Therefore, resource allocation algorithm must consider the latency of subtasks waiting for their predecessors completed. Subsequently, we give the definition of ready time of a subtask [22].

The ready time of a subtask is defined as the earliest time when all immediate predecessors of the subtask have completed execution. Therefore, the ready time of subtask *m* which executed on server *w* is given by
(5)$$RT_{m}=\max_{k\in pred(m)}\left\{RT_{k}+T_{k,j}^{exc}+T_{k,j,w}^{trans}\right\},$$
where pred(m) denotes the set of immediate predecessors of subtask *m*.

### 2.4. Resource Allocation Problem Formulation

In this section, we formulated the resource allocation problem that was subject to several constraints. Each task *t* of an environmental monitoring application *a* can be denoted as $t=\{T,t_{deadline}\}$, where T indicates a set of subtask correspond with modules *M* of *a*. To ensure the Quality of Service (QoS) of environmental monitoring application, task *t* should be completed before $t_{deadline}$. In terms of the limited computation resource of edge computing servers, we consider the resource allocation problem: decide which server should execute the subtask $t_{i}\in T$. We defined the integer decision variable $x_{i}\in S$ that indicates subtask $t_{i}$ executed on server $x_{i}$, where S represents a set of servers that can serve the region where the sensor located. The variable of resource allocation is as follows: $X=\{x_{1},x_{2},\ldots ,x_{m}\}$. We propose minimizing the task completion latency $T_{comp}$.

Formally, the resource allocation problem can be formulated, as follows:(6)$$\begin{array}{cc} \underset{X}{\mathrm{minimize}} & \;\max_{k\in T}\left\{RT_{k}\right\}+T_{k,x_{k}}^{c} \end{array}$$(7)$$\begin{array}{cc} \mathrm{subject}\;\mathrm{to}: & \;\forall i\in S,\forall j\in T, \end{array}$$(8)$$\begin{array}{cc} \; & \;T_{comp}\leq t_{deadline}, \end{array}$$(9)$$\begin{array}{cc} \; & \;flag_{x_{i}}\geq level_{j},j\in T, \end{array}$$(10)$$\begin{array}{cc} \; & \;ST_{j}\geq \max_{k\in pred(j)}\left\{RT_{k}+T_{k,s}^{exc}+T_{k,s,x_{j}}^{trans},\right\} \end{array}$$

Constraint (8) is a completion time constraint that restricts the total completion time of all the subtasks of an environmental monitoring application that is limited by the required maximum completion time $t_{deadline}$, which is determined by the type of application. The deadline of routine monitoring application (e.g., generating pollution map) is typically larger than emergency applications (e.g., pollution warning). Constraint (9) states subtask $t_{j}$ can only be executed on a certain devices on $layer_{j}$ or above. $ST_{j}$ in constraint (10) describes that the start time of subtask *j* is not earlier than the latest time of its predecessors executed and transmitted to server $x_{j}$.

## 3. Resource Allocation Algorithm and Task Scheduling Strategy

As aforementioned, to reduce average task completion time, the resource allocation algorithm and task scheduling policy need to consider the dependencies among subtasks. In this section, an improved GA based resource allocation algorithm and a task scheduling policy were proposed. When a task is offloaded to the edge computing network, the server implements the resource allocation algorithm for this task, and outputs a resource allocation policy that specifies the processing server of each subtask. During the operation of the edge computing system, each edge server periodically sorts the task queue by implementing the proposed task scheduling strategy.

### 3.1. GA Based Resource Allocation Algorithm

In term of the cost of genetic algorithm is positively correlated with the amount of subtasks and specific subtasks of environmental monitoring application can only be executed on certain servers. In this paper, we propose a suboptimal algorithm, named combined genetic algorithm (CGA), as shown in Algorithm 1.

#### The Flow of CGA

A number of works use the Genetic Algorithm to deal with the resource allocation problem. Genetic algorithm adopts the idea of survival of the fittest from the theory of evolution. It performs crossover and mutation operations on a population of solutions until reaching the optimal solution. However, with the increase in the number of subtasks to be allocated, the time cost of genetic algorithm increases rapidly. In addition, resource allocation policies tend to assign more dependent subtasks to the same server. If two subtasks with heavy dependency are divided to be executed on the same server, the communication latency between the two subtasks can be ignored. In addition, calculating the resource allocation result also consumes the resources of edge computing system, and it is usually calculated on roadside server with fewer computing resource. Based on this characteristic, this paper uses the k-means clustering method to cluster subtasks, and take each subtask group as the smallest unit of resource allocation, in order to reduce the size of the solution space. CGA in this work can be described, as follows:

*(1) Clustering subtasks:* the k-means clustering algorithm first selects the *k* subtask as initial centroids. Subsequently, it calculates the dependencies among subtasks, and adds a subtask with the heaviest dependency with a certain centroid into the its subtask group. Next, k-means recalculates the centroids of each subtask group and repeats the previous operations until the centroids are stable, as shown in Algorithms 2 and 3. To cluster subtask of an application, we consider divided subtasks with heavy communication dependency into a same group. By clustering $\left\{t_{1},\;t_{2},\;t_{3}\right\},\;\left\{t_{4},\;t_{5},\;t_{7}\right\}$, each subtask cluster can be the smallest unit in resource allocation algorithm, as shown in Figure 3.

By grouping the subtasks, the cost of genetic algorithm and the communication among subtasks can be reduced. However, subtask grouping makes us consider the feasibility of solution in the generation initial population, crossover, and mutation operations. Because a certain subtask in a group can only be executed on specific server, the whole group should be scheduled to specific server, which can be described, as follows:(11)$$flag_{x}\geq \max\left\{layer_{i}\right\},i\in G_{j},$$
where $G_{j}$ denotes a subtask group and $flag_{x}$ denotes the server layer of solution of group $G_{j}$.

CGA optimizes a set of initial random solutions to a acceptable solution, by evolutionary operations (selection, crossover, and mutation). The crossover and mutation operations of CGA can maintain solutions keep diversity avoid falling into local optimal trap.

*(2) Chromosome and Fitness Function:* in this paper, we use real coded GA. Each individual is defined by the chromosome, which describes a solution of problem (6). Figure 4 shows the chromosome structure of an individual. $x_{i}$ in chromosome structure denotes a target server that is assigned to subtask cluster $c_{i}$.

To evaluate the feasibility of an individual, the fitness function is defined, as follows:(12)$$Fitness\left(X\right)=\max_{i\in ot}\left\{RT_{i}+T_{i,x_{i}}^{c}\right\},$$
where ot is a set of subtask without outdegree in task DAG. It describes that the end task of an application.

*(3) Initialization and Selection:* as shown in Algorithm 1, first get the subtask cluster of an environmental monitoring Application *a* by Algorithm 2. Subsequently, generate the initial solution population randomly, but constrained by (11). The selection operation of CGA restricts partially individuals that fare better than others can survive.
**Algorithm 1:** Get subtasks cluster.
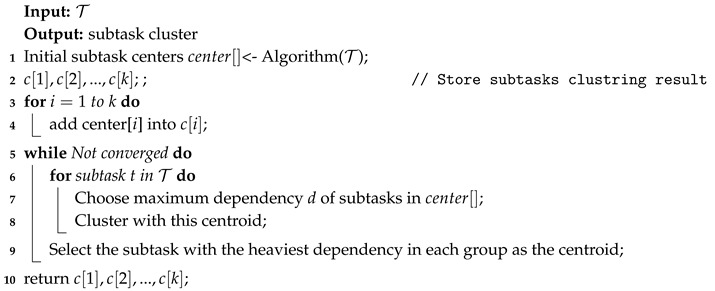

**Algorithm 2:** Get initail subtask center.
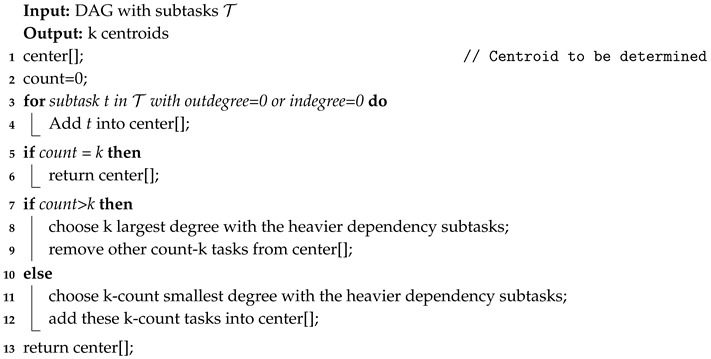

**Algorithm 3:** CGA.
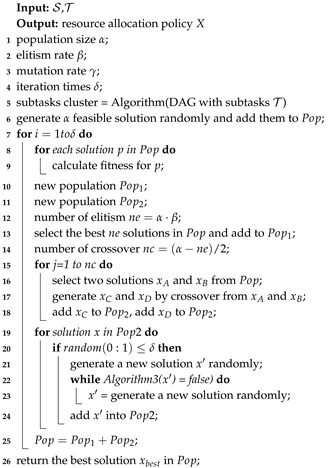


*(4) Crossover and Mutation:* to maintain the diversity of population and reach a better solution to resource allocation problem, CGA applied crossover and mutation operations iteratively to the population. In each iteration, the selection chooses suitable individuals for crossover operation. Subsequently, the crossover operation is applied to these individual in order to generate their offsprings. Because subtasks are already clustered, in this work we only consider crossover one bit in a chromosome, in order to maintian local search capability. As shown in Figure 5, crossover operation select one bit randomly from chromosome of Parent α and Parent β, and generates offspring α and offspring β. Because the crossover operation switch the same index on chromosome that indicates a same subtask group, the solutions of offsprings generated by crossover are also feasible.

To avoid the population falling into local optimal trap, the mutation operation alter chromosome of individual randomly, but constrained by (11).

### 3.2. Task Scheduling Strategy

As aforementioned in Section 2, the ready time is an important factor of the completion time of a task. Additionally, the earliest start time of a subtask is when the predecessors of this subtask completed and transmitted to the server that this subtask assigned. Clearly, reducing the completion time of the latest predecessor task could reduce the completion time of the whole task. However, the ready time and deadline of task are ignored by traditional first come first service (FCFS) task shceduling strategy. In order to reduce the cost of ready time and take deadline into account, we propose a task scheduling strategy shown as Algorithm 4. When a subtask is completed, the server records the subtask Id into completion table that maintained on each server. Additionally, servers sort the waiting queue periodically according to the priorities of subtasks in waiting queue. The sorting operation follows that, if a subtask being executed earlier could reduce the completion time of whole task, it should be served in advance.

Subtasks $t_{1},t_{2},t_{3}$ and $t_{1}^{*}$ with green describes these subtask has been executed, as shown in Figure 6. Therefore, serving $t_{4}$ ahead of time can fully use the parallel computing capacity of edge computing system to reduce the waiting time.

In addition, the proposed task scheduling strategy also tends to serve the emergency task. The strategy sort the task queue on server according to priority *p* of subtask, which can be calculated by Equation (Equation 13).
(13)$$p=\alpha {\left(t_{deadline}-t_{cur}\right)}^{-1}\cdot c,$$
where $t_{deadline}$ denotes the deadline of the task that depends on the type of environmental monitoring application. $t_{c}ur$ describes Current system time. *c* describes the number of completed subtasks in the set of peer subtasks, which denoted by pt in Figure 6. Peer subtasks are defined as the predecessors of successors of $t_{i}$, which can be described, as follows:(14)$$\forall t_{j}\in pt,\;\exists d^{{}^{\prime }}=\left\{m_{i},m_{k}\right\}\;and\;d^{{}^{”}}=\left\{m_{j},m_{k}\right\},$$
where $m_{i}$ and $m_{j}$ denote the application modules of $t_{i}$ and $t_{j}$ in DAG, respectively. Because resource allocation results are carried by data packages in edge network. The task completion record can be requested quickly according to the allocation result, even if it is not maintained on the current server.
**Algorithm 4:** Task scheduling strategy.
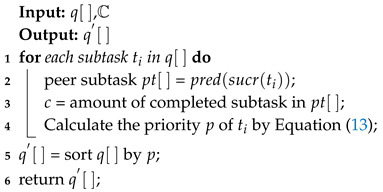


## 4. Simulation and Result

### 4.1. Experimental Environment

In this work, we extend iFogsim to support the architecture in Section 2. The Tuple class is modified to implement successors to inherit the resource allocation result and task deadline. The modified FogDevice class communication with other devices located on a same layer to implement collaborative computing and load balancing. The resource allocation algorithm interface is placed inside the FogDevice.

Servers settings: environmental monitoring sensors are installed on roadside monitoring station or vehicles, and connected to roadside edge server. The CPU capacity of each roadside edge server is in the range of 3000–6000, and the Ram range of 4000–8000. The CPU capacity, Ram, and bandwidth of region server are 8000–12,000, 20,000, and 10,000, respectively. We set the CPU capacity of cloud server as 44,000, the Ram as 40,000, and the bandwidth as 10,000. Additionally, we set two type of connection between end device (monitoring station and vehicles) and edge server: fixed number, random number. The detailed settings are shown in Table 1.

After setting up the network architecture and devices on each layer. The application model is set, as shown in Figure 7.

The mapping relationship between application modules and servers on different layers are shown in Table 2.

### 4.2. Simulation Result

In this section, we evaluate the performance of the proposed method in the iFogsim simulation environment. We compare our method with the basic genetic algorithm to prove the effect of our resource allocation algorithm and task scheduling strategy. In our experiment, we first observe the convergence of GA and CGA in a same time of resource allocation. Subsequently, we compare the average task completion latency with different sensor amount deployed on a roadside server, in order to ascertain a suitable number of sensor per roadside server to serve. In addition, the simulation result of average task latency with a different amount of request per sensor offload in unit time prove the minimum time granularity for environmental monitoring in this experimental environment. In the following results, GA represents basic Genetic Algorithm, CGA denotes the Combined Genetic Algorithm we proposed, and combined genetic algorithm with task scheduling strategy is denoted by CGA&Q.

### 4.3. Convergence Analysis

Figure 8 shows the different convergence between GA and CGA. On account of grouping subtasks, CGA converged earlier than GA in earlier iterations, but GA could reach a better sollution than CGA in later iterations. However, consider the roadside server’s lack of computing resource, the iteration is set to the range 50–200 times in this work. Obviously, in this range, the CGA we proposed could reach a better sollution than GA.

### 4.4. Performance Analysis

In this section, we first show the performance of proposed algorithm versus GA with different sensor amount under a roadside server. Figure 9 plots the average task completion latency as the result in different methods. Each sensor offloads 16 environmental monitoring tasks in a time unit. When less than seven sensors are deployed on a roadside server, the effect of CGA&Q is not obvious, due to the size of task queue on each server being small. With the increase of sensor number, the performance of proposed method becomes higher. In the ten scenarios, the average completion latency optimization ratio (GA vs. CGA&Q) is 14.1%, 10.0%, 10.5%, 9.8%, 7.6%, 13.6%, 12.0%, 22.4%, 16.4%, and 13.0%. As the sensor number grows above 10, it has been hard for completion latency to meet the demand of environmental monitoring application. Therefore, in following simulation, we set the number of sensors that are deployed on roadside server to 10.

Figure 10 shows the average completion latency with different task number per sensor offloaded in a unit time. In the eight scenarios of different task amount, the average completion latency optimization ratio (GA vs. CGA&Q) is 18.4%, 16.4%, 15.5%, 18.1%, 21.6%, 21.1%, 18.7%, and 6.9%. Because each bit in the chromosome only denotes one subtask, the sollutions in the initail population of GA will cost more time on communication than CGA. When we decrease the number to 8, the performance of CGA is better than CGA&Q. Because the priority of subtask in each task queue are non-real time (behind the current system state). As task number decreases, the task queue refresh speed increases. It degrades the performance of the task scheduling strategy.

To ascertain the ratio of subtasks number executed by the different layer servers, We deploy 10 sensors on each roadside server, and set region server connected with eight roadside servers. Figure 11 summarizes the utilization of server on different level. As the amount of tasks increases, the proportion of tasks allocated to cloud computing increases from 14.06% to 31.41%. Meanwhile, the average completion latency increased from 1458 ms to 2910 ms, because more subtasks were allocated to the cloud server. Even if the less task offloaded into edge computing system, the region server and cloud server still execute the task partially, because a certain specific subtask can only be executed on the cloud or region server.

## 5. Conclusions

This paper takes the characteristics of environmental monitoring applications into account, and presents a resource allocation algorithm and task scheduling strategy for an edge-computing based environmental monitoring system. By clustering subtasks with heavy communication dependency, the proposed method reduces the cost of calculating results of resource allocation and accelerates the convergence of the genetic algorithm in the early iterations. In addition, we propose a task scheduling strategy to fully use the parallel computing capacity of edge computing system. Subsequently, we ascertain the suitable setting of environmental monitoring system in the environment of this work. However, the clustering subtask should be dynamically adjusted to the status (i.e., real-time resources and load of edge servers) of edge computing system. Additionally, the additional communication cost (e.g., if completion information are not on current server, it will be transmitted by edge network) caused by task scheduling strategy has not been investigated. In future work, we plan to improve the proposed methods to adjust the allocation policy dynamically, and implement a real-world testbed in order to improve the proposed method and system model in terms of cost of resources and availability of monitoring service.

## Figures and Tables

**Figure 1 sensors-20-06125-f001:**
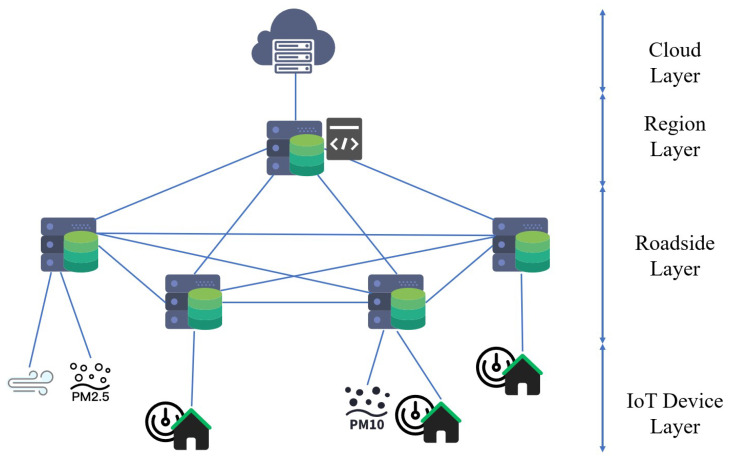
Edge-computing based environmental monitoring system model.

**Figure 2 sensors-20-06125-f002:**
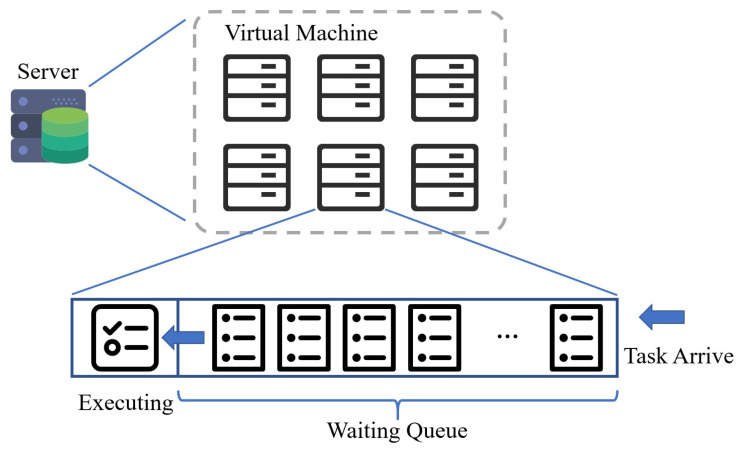
Server Model.

**Figure 3 sensors-20-06125-f003:**
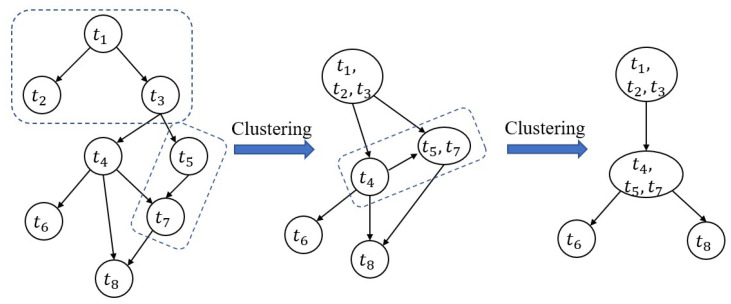
Clustering substaks.

**Figure 4 sensors-20-06125-f004:**
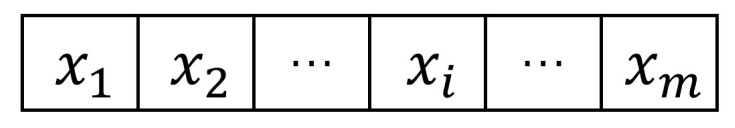
The chromosome structure.

**Figure 5 sensors-20-06125-f005:**
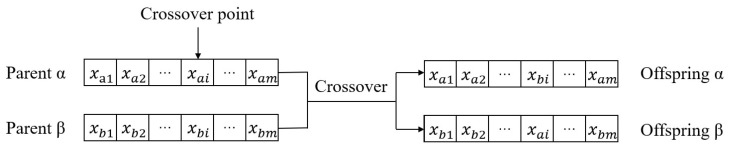
Crossover operation.

**Figure 6 sensors-20-06125-f006:**
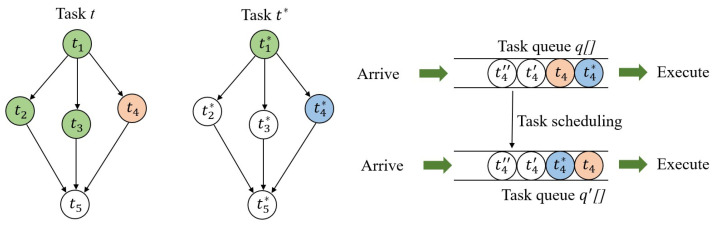
Task scheduling.

**Figure 7 sensors-20-06125-f007:**
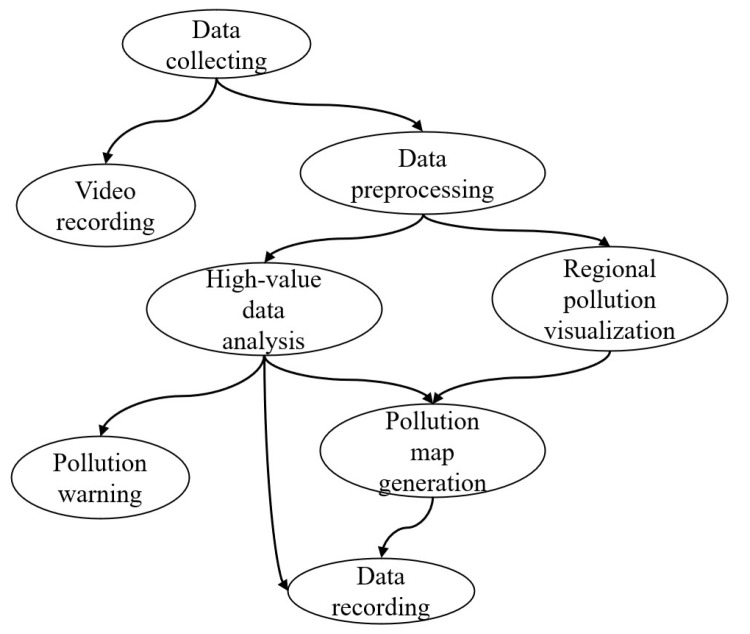
Environmental monitoring application modules.

**Figure 8 sensors-20-06125-f008:**
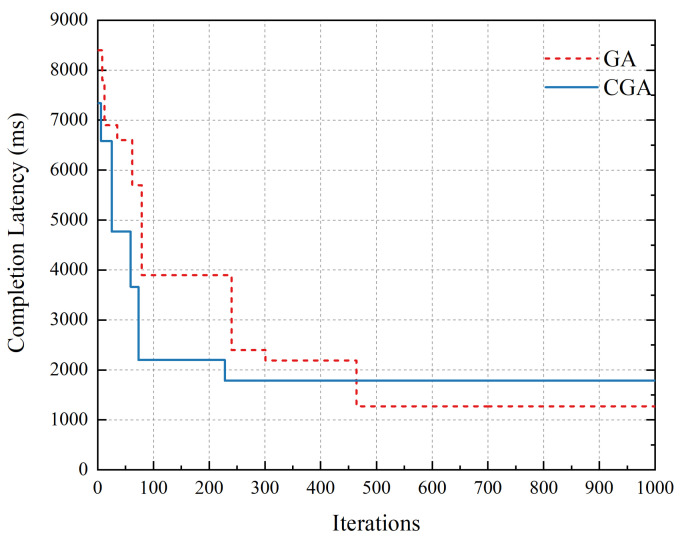
Convergence of GA vs. CGA.

**Figure 9 sensors-20-06125-f009:**
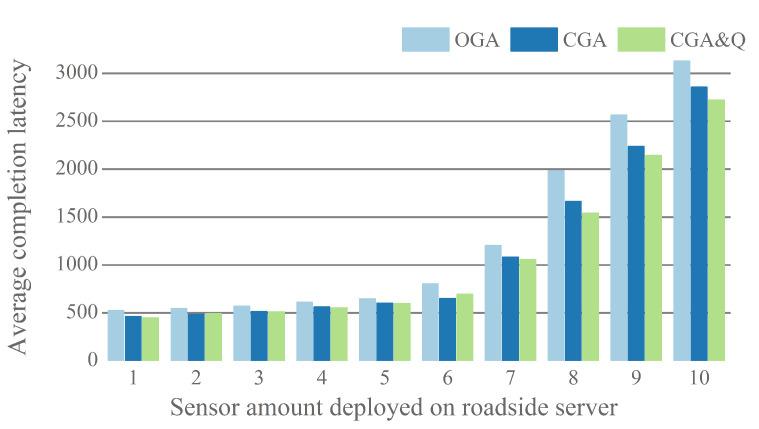
Impact of sensor amount on average completion latency.

**Figure 10 sensors-20-06125-f010:**
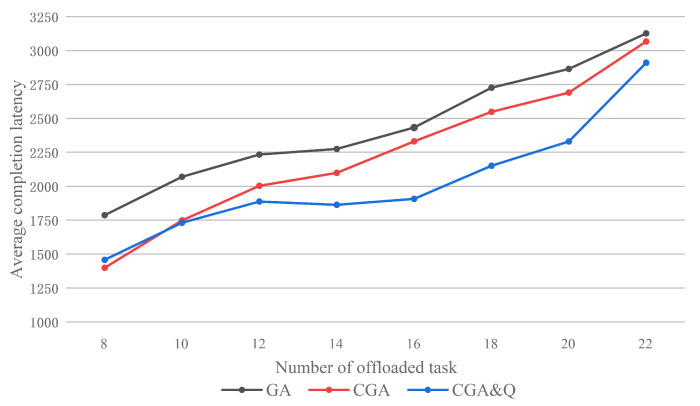
Average completion latency of application by GA, CGA, and CGA with task scheduling.

**Figure 11 sensors-20-06125-f011:**
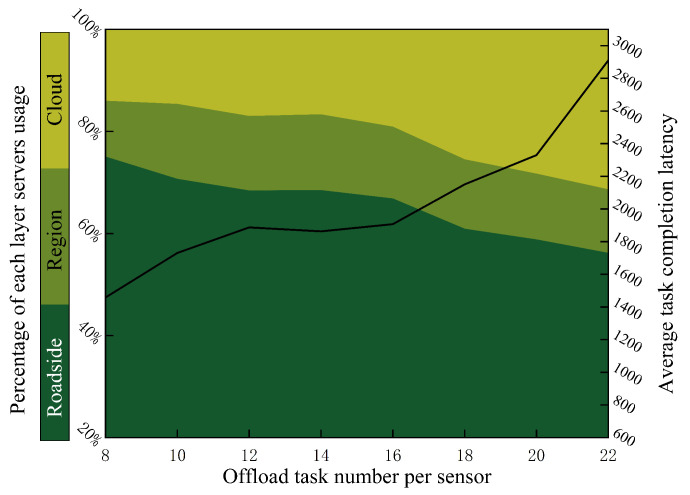
The utilization of servers in each layer.

**Table 1 sensors-20-06125-t001:** Experimental simulation parameter settings for servers.

	CPU (MIPS)	RAM (MB)	Up/Down Bandwidth (bytes/ms)	Busy/Idle Power (W)
Cloud server	44,000	40,000	10,000/10,000	1648/1332
Region server	8000–12,000	8000	10,000/10,000	107.34/83.43
Roadside server	3000–6000	4000–8000	5000/10,000	87.43/60.43

**Table 2 sensors-20-06125-t002:** Mapping between application modules and servers.

Layer	Application Module
Cloud	Data recording
	Pollution map generation
	Regional pollution visualization
	High-value data analysis
	Vedio recording
Region	Data preprocessing
	High-value data analysis
	Regional pollution visualization
	Vedio recording
Roadside	Data preprocessing
	High-value data analysis
	Vedio recording
IoT Device	Data collecting
	Pollution warning

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
