# Peer review of "An Efficient Resource Allocation Strategy for Edge-Computing Based Environmental Monitoring System"

_sensors, 2020, doi:10.3390/s20216125_

Round 1

Reviewer 1 Report

The authors propose a resource allocation method for edge computing.

This is a hot topic of research. Therefore, several solutions have been proposed recently. The works mentioned in the related work are recent but there are several other important works published in this year and in the last year that must be referenced and compared with. Please take a look at some of the works found in IEEE.

The description of previous works is too succinct. There should be a qualitative comparison and a clear description of why the proposed work is better and in what aspects.

Some sentences must be justified with at least a reference. For example, in line 61 “… consume too much computing resource of edge computing server.”

In section 2.1 you mention a network architecture. Are assuming this architecture in our models? Is it important to consider this architecture? Do your proposed methods depend on this architecture being followed? I am just trying to understand why this network architecture and not another.

The clustering considers k sub tasks. How do you determine this value?

In your report, you do not mention the processing flow of allocation/scheduling algorithms and execution of tasks. It would be important to understand when the allocation and scheduling algorithm run. How often do you execute it? Do you run it whenever new tasks need to run?

In the experimental environment (section 4.1), there should be a reference to the table. Also, the figures in the table should be explained. What are the units? Why these values?

In section 4.3, about the analysis of converge, you consider an iteration range of 200-500. Why? Also, it is important to show the execution times of both algorithms. This way we can take conclusions about the feasible number of iterations. Also, it also considers one case study. Can you take conclusions using one case study?

Performance analysis is based on the fact that we are adjusting the number of iterations to between 200-500. Otherwise, GA would be better, that is, would found a better solution. So, it is important to clearly justify this range of acceptable iterations of the algorithms.

The plot in figure 11 is some difficult to read. What you are telling is that when the number of tasks increase a higher percentage of tasks are sent to the region or cloud. Is it true?

A final note goes the written English. A careful revision must be done to improve the readability of the document.

Reviewer 2 Report

To reduce the average completion latency of environmental monitoring application, this paepr proposes a resource allocation algorithm based on genetic algorithm and then introduces a task scheduling strategy. The topic is interesting and improtant. Although there are lots of previous studies, this paper has some new ideas and contributions.

The following comments can help the authors improve the quality of this paper.

1. Introduction has to be improved to clearly present the motivation of this paper and challenges of the related studies. The contributions are also summarized explicitly. Current version cannot introduce the detailed contributions of this paper. For example, k-means clustering method used to cluster subtasks is not mentioned at all.

2. The authors should add a notation table since there are so many variables in this paper. Besides, the table in page 10 of subsection 4.1 should have a caption.

3. This paper did not give a good review about the related works. There are lots of previous works about IoT and edge computing, such as
[1] Joint Optimization of Energy Consumption and Latency in Mobile Edge Computing for Internet of Things. IEEE Internet of Things Journal, Vol. 6, Issue 3, June 2019, pp. 4791-4803.
[2] Data-Aware Task Allocation for Achieving Low Latency in Collaborative Edge Computing. IEEE Internet Things J. 6(2): 3512-3524 (2019)

4. I think Section 3 is not well organized. In the beginning, the pseudocode of proposed combined genetic algorithm (CGA) is given. In this part, the readers cannot understand this proposed algorithm. I think the pseudocode should be placed at the end of subsection 3.1.

5. I want to know why the GA algorithm is chosen since there are so many evolutionary algorithms, such as ABC and DE. The authors should give an explaination or give some experimental results.

6. Regarding clustering subtasks, the dependencies among subtasks are calculated, and add subtask with heaviest dependency with a certain centroid into the its subtask group. How to define "dependency" and how to calculate it should be given the details. This is a key issue for clustering and the clustering effect is important for CGA.

7. In the experiments, there are some problems. Regarding convergence, current experimental design is not reasonable. I think the time reaching the optimal solution should be compared. To validate the effectiveness of clustering subtasks, CGA should be compared with CGA without k-means.

8. There are so many grammatical mistakes in this paper. The authors should carefully improve the quality of linguistics.

Round 2

Reviewer 1 Report

Technically the contents are fine. However, I recommend a revision of the English language.

Reviewer 2 Report

The authors have addressed all my concerns. Thus, I recommend to accept this paper.